# Mediating factors in the relationship between combat-related traumatic injury and myocardial blood flow reserve: The ADVANCE cohort study

**Christopher J Boos**[1,2,3,4]*, **Susie Schofield**[5], **Rabeea Maqsood**[3], **Anthony MJ Bull**[3], **Nicola T Fear**[1], **Paul Cullinan**[6], **Harriet Kemp**[7], **Alexander N Bennett**[3,8], for the ADVANCE Study¶

1 Academic Department of Military Mental Health and King's Centre for Military Health Research, King's College London, London, England, 2 Faculty of Health & Social Sciences, Bournemouth University, Bournemouth, England, 3 Centre for Injury Studies, Department of Bioengineering, Imperial College London, London, England, 4 Department of Cardiology, University Hospitals Dorset, Poole Hospital, Poole, England, 5 National Heart and Lung Institute, Faculty of Medicine, Imperial College London, London, England, 6 London, United Kingdom, 7 Department of Surgery & Cancer, Imperial College London, London, England, 8 Academic Department of Military Rehabilitation, Defence Medical Rehabilitation Centre, Stanford Hall Estate, Loughborough, Nottinghamshire

¶ Membership of the ADVANCE Study is listed in the Acknowledgments.
* s.schofield@imperial.ac.uk

## Abstract

### Introduction

Combat-related traumatic injury (CRTI) has been linked to reduced myocardial blood-flow- reserve (MBFR), measured using the subendocardial viability ratio (SEVR). We aimed to assess the mediating role of known cardiovascular risk factors on SEVR.

### Materials and Methods

We examined 1018 UK servicemen (prospective ADVANCE Cohort Study) comprising 504 with CRTI (140 amputees) and 514 uninjured men, frequency-matched at sampling, by age, rank and deployment (Afghanistan 2003–2014). We examined the mediating role of cardiovascular risk factors, shown to significantly greater with CRTI at study baseline (~8 years post-injury/deployment), on SEVR, measured three-years later (FU1). The cardiovascular risk measures were heart-rate variability (HRV, root-mean-square-of-successive-differences [RMSSD]), visceral-fat-mass (VFM, using DEXA), venous-blood high-sensitivity C-reactive protein (Hs-CRP, inflammation), six-minute walk distance (6MWD, physical function) and weekly leisure-time moderate-to-vigorous physical activity (LT-MVPA, physical activity).

### Results

At baseline, VFM was significantly greater and RMSSD, 6MWD and LT-MVPA lower with CRTI compared to the uninjured. VFM and Hs-CRP were significantly greater

**Data availability statement:** The data underlying this study are not publicly available due to ethical and legal restrictions. Although the dataset has been de-identified, it contains sensitive personal and clinical information. The combination of these variables poses a non-trivial risk of participant re-identification, particularly for individuals in small or vulnerable subgroups. For this reason, unrestricted public sharing of the data could compromise participant confidentiality. These restrictions were imposed by the ADVANCE Study Project Board to protect participant confidentiality. Data may be made available upon reasonable request to qualified researchers, subject to approval by the relevant committee and the implementation of appropriate data protection safeguards. Requests should be directed to the ADVANCE study Data Team at: adv_data_team@imperial.ac.uk.

**Funding:** This study is a project by the ADVANCE study (grant number: ADV-ADMR-03), which is funded through the ADVANCE Charity. Key contributors to the charity are the Headley Court Charity (principal funder), HM Treasury (LIBOR Grant), Help for Heroes, Nuffield Trust for the Forces of the Crown, Forces in Mind Trust, National Lottery Community Fund, Blesma - The Limbless Veterans, the UK Ministry of Defence, and the Office for Veterans' Affairs (OVA). The funders of the study had no role in study design, data collection, data analysis, data interpretation, or writing of the manuscript.

**Competing interests:** A.N.B is a serving member of the Royal Air Force. NTF is a trustee of Help for Heroes and is part-funded by a grant from the UK Ministry of Defence. All the remaining authors declare no conflicts of interest. The views expressed are those of the authors and not necessarily those of the UK Ministry of Defence.

and RMSSD, 6MWD and LT-MVPA lower in the injured amputees versus the injured non-amputees and uninjured. The SEVR at FU1 was significantly lower in the injured (mean ± standard deviation; 187.2 ± 39.7) compared to the uninjured (194.1 ± 31.5) and lowest in the amputee sub-group (181.9 ± 30.0). The association between CRTI and SEVR was mediated by VFM (natural indirect effect −1.80: 95%CI: −2.90, −0.67), RMSSD (−1.82: −3.45, −0.19) and 6MWD (−1.79: −3.16, −0.41) but not Hs-CRP and LT-MVPA. The association between traumatic amputation and SEVR was mediated by VFM, HRV and LT-MVPA.

## Conclusions

VFM, HRV and physical function/activity were significant mediators of the link between CRTI and SEVR. Interventions on physical activity/function could mitigate the association between CRTI and SEVR. Data from longer-term follow up are required to robustly determine the temporal effects of this relationship.

## Introduction

There is an increasing body of evidence to support an association between combat related traumatic injury (CRTI) and elevated cardiovascular risk [1–3]. Whilst the majority of the supportive data have been derived from retrospective observational studies relating to historical conflicts, the recent wars in Iraq and Afghanistan have led to more robust and contemporaneous evidence [4,5].

Recently published baseline data from the ArmeD SerVices TrAuma and RehabilitatioN OutComE (ADVANCE) Cohort Study have demonstrated a significant relationship between CRTI and increased cardiovascular risk, quantified using established cardiovascular risk measures [4,6]. In ADVANCE, the participants with CRTI had significantly greater abdominal obesity, visceral fat and high-sensitivity C-reactive protein (Hs-CRP) and lower physical function, activity and heart rate variability (HRV, a measure of autonomic balance) compared with a frequency-matched group of uninjured combatants exposed to the same conflict and deployed at a similar time [4,6,7]. Moreover, the risk was greater with worsening injury severity and following traumatic limb amputation [4,6,7].

One of the potential mechanisms for this increased risk appears to be a reduction in relative myocardial blood flow reserve (MBFR), measured using the subendocardial viability ratio (SEVR) [7]. The SEVR, also known as the diastolic time index/systolic time index ratio, is a hemodynamic marker that reflects the balance between myocardial oxygen supply (primarily to the sub-endocardium) and myocardial oxygen demand [8,9]. Lower SEVR has been strongly linked to reduced coronary microcirculatory function and increased cardiovascular risk and adverse cardiovascular outcomes [10–12].

Baseline data from the ADVANCE Study, measured at an average of eight years post CRTI, demonstrated that CRTI and its worsening severity was independently associated with lower SEVR [7]. Whilst association does not equate to causation,

the statistical method of mediation analysis can provide a better understanding of the direct and indirect effects of CRTI on coronary blood flow and the potential mechanisms through other pathways to explain their relationship. Moreover, by utilising longitudinal methods, causal inferences can be explored.

In this study, we sought to clarify the longitudinal relationship between CRTI and SEVR by investigating potential mediators of the relationship using data from multiple phases of the ADVANCE Study. We examined the relationship between key baseline cardiovascular risk measures, noted to be greater in the injured versus uninjured, were compared with the SEVR measured at the first study follow up three years later.

## Methods

### Study population and design

The study population were participants recruited into the on-going ADVANCE Prospective Longitudinal Cohort study. For this study we examined 1018 participants from the ADVANCE Study who had both baseline and three-year follow SEVR data available. This consisted of 514 adult (≥18 years) UK military servicemen who sustained CRTI (sufficient to require aeromedical evacuation and admission to UK hospital) during combat operations in Afghanistan (2003–2014) and 504 uninjured servicemen who were frequency-matched, at sampling, to the injured based on age, rank, sex (male), regiment, role in the theatre, and deployment period [13]. Key exclusion criteria were any participants with established cardiovascular disease prior to their exposure (injury/deployment of interest) or evidence of active acute infection.

Participation was voluntary and followed full written informed consent. Ethical approval was granted by the MoD Research and Medical Ethics Committee (MoDREC: 357PPE12) and the study complies with the principles outlined in the Declaration of Helsinki. The first patient was recruited on the 1st August 2015 and the three-year follow-up ended on the 31st August 2024.

### Data collection

All clinical data were collected at a baseline study visit, conducted at an average of eight years post exposure (injury/deployment), and again three years later at their first planned follow up visit. The detailed description of the baseline cohort and their demographics have been previously described and published [4,13].

Participants were asked to fast and refrain from caffeine and alcohol for at least eight hours and smoking for ≥4 hours prior to their study visit. All venous blood tests which were sent to the local NHS laboratory for blind processing. Collected information and demographics included confirmation of the participant's ethnicity, medical, family (of stroke or coronary heart disease and smoking histories. Military rank was classified into: senior rank (commissioned officers), mid rank (senior non-commissioned officers) and junior rank (junior non-commissioned officers and other lower ranks) [4]. Injury severity was quantified using the New Injury Severity Score (NISS) [13]. The NISS ranges from 1–75 and is calculated as the sum of the squares of the highest Abbreviated Injury Scale grade in each of the three most severely injured body regions (head/neck, face, thorax, abdomen, extremities, and external) [14].

### Baseline measures

Vascular inflammation was measured using venous blood Hs-CRP. Visceral fat mass (VFM, kg) was quantified using dual-energy X-ray absorptiometry (DEXA, Vertec Horizon and Discovery, UK) [13]. Physical activity was calculated using the long-form of the International Physical Activity Questionnaire (IPAQ) [15,16]. The total weekly burden of leisure-time moderate to vigorous physical activity (LT-MVPA) was quantified as the total weekly minutes of LT-MVPA with each time variable truncated to a maximum of 180 minutes per day for each modality as previously described [15]. The total weekly LT-MVPA was graded according to the World Health Organisation (WHO) weekly recommendation of ≥150 minutes of moderate and/or ≥75 minutes of vigorous exercise [17–19]. Physical-function was measured using the six-minute walk test (6MWD). The primary HRV measure was the root mean square of successive differences (RMSSD), which was

processed using Kubios Premium HRV software. It was calculated using consecutive inter-beat-interval data obtained from an ultrashort 14 second analysis of the femoral arterial pulse waveform using the Vicorder device (Skidmore Medical, UK) as previously described and validated [6,20].

### Primary outcome – Subendocardial Viability ratio *at the* Three Year Follow *up*

The SEVR was measured at the three-year follow up using brachial artery pulse waveform analysis (Vicorder® device) as previously described and validated [7,21]. As at baseline, the measurements were undertaken by trained research nurses in a temperature-controlled room after the participants had rested for five minutes, lying supine with their head raised to 30° [4]. The SEVR was automatically calculated by the Vicorder® software as the diastolic pressure-time index (DPTI) divided by systolic pressure time index (SPTI). Diastolic pressure-time index is the area between aortic and left ventricular end-diastolic pressure (LVEDP) curves during diastole and reflects coronary perfusion (myocardial oxygen supply). The systolic pressure-time index is the area under aortic pressure curve during systolic ejection period (SEP) and reflects myocardial oxygen demand, which increases with systolic workload [11]. A lower SEVR suggests reduced oxygen supply relative to demand, and reduced MBFR and possible subendocardial ischemia. All Vicorder measures were done in triplicate, and the median value was used. If the coefficient of variation (CV) of the three values was ≥ 30%, the median of the two values with CV < 30% was used. One participant had a CV ≥ 30% for all values and was excluded.

Reproducibility was excellent with the median coefficient of variation for ≥3 repeat SEVR readings being 4.96% (interquartile range [IQR] 2.92–8.11%).

### Statistical analysis

Data were analysed using the statistical software package Stata version 18.0 [22].

Of the 1053 participants who returned for the first follow up, there was missing SEVR data for 35 participants, leaving 1018 for these analyses. We also had missing data on the individual mediators: 33 for 6MWD, 55 for Hs-CRP, 253 for RMSSD (unavailability of raw waveform data) and 48 for LT-MVPA.

Continuous data were presented as means (standard deviations [SD]), or, where their distribution is skewed, by medians (interquartile ranges, IQR). Unpaired T tests or Mann-Whitney U tests, as appropriate, were used in two-group comparisons of continuous data; three-group comparisons were made using one-way ANOVA or Kruskal-Wallis tests. The Chi-squared or Fisher's exact tests were used to compare categorical data. Spearman's rank correlation was used to assess the correlation between the mediators. We defined the strength of correlation as a rho (ρ) <0.2 indicating no/very weak correlation, 0.2–0.4 weak correlation and >0.4 moderate to strong correlation [23].

To examine the extent to which the association between CRTI and SEVR is mediated by VFM, physical activity (LT-MVPA) and function (6MWD), HRV (RMSSD) and inflammation (Hs-CRP), we used counterfactual mediation methods (Fig 1) [24]. The STATA package *mediate* was used to decompose the effects into natural indirect effects (NIE), natural direct effects (NDE) and total effects (TE) and are reported as unstandardized coefficients and their 95% confidence intervals (CI). An exposure-mediator interaction was included [25]. We assessed each mediator separately. A priori confounders (age at injury/deployment and rank) of the exposure-mediator and mediator-outcome relationships were included in the models. Proportion mediated (PM) was calculated as $PM = \frac{TE-NDE}{TE}$. We also reported the controlled direct effect (CDE) for potentially modifiable mediators, which was defined as the effect of the exposure on an outcome that is observed if the mediator were fixed at a certain level and is the effect due to neither mediation nor interaction. Additionally, we reported the proportion eliminated (PE), calculated as $PE = (TE - CDE)/TE$ [26]. To assess the assumption of normality of residuals of the mediation model, a Q-Q plot of residuals (based on a linear regression model of the association between exposure and mediator with confounders) was inspected. Bias-corrected bootstrap confidence intervals using 1000 replications are reported to account for any deviations from residual normality. The analysis was repeated for the subgroups of amputation status (uninjured (reference), injured non-amputation, injured amputation).

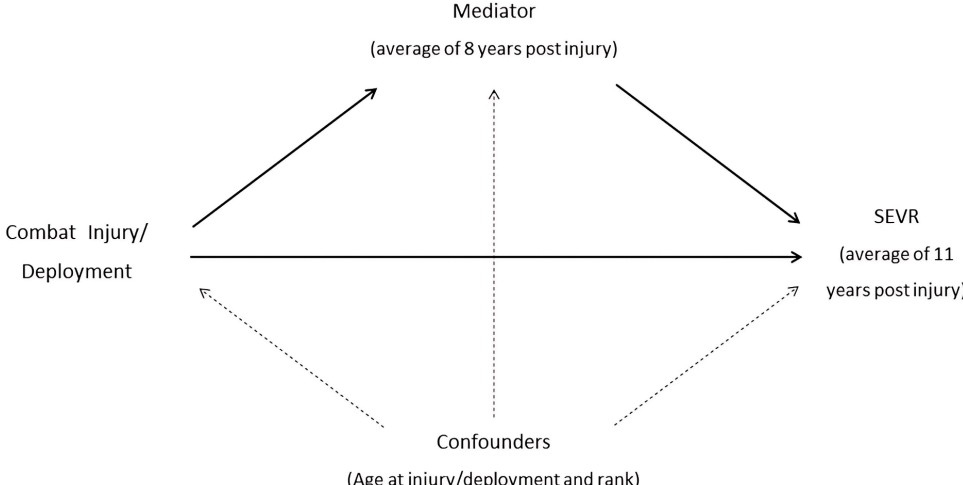

**Fig 1. Theoretical mediation model.**

## Results

Our cohort consisted of 1018 participants with measurements of SEVR at the three-year follow up visit comprising 504 men with CRTI and 514 who were uninjured (Table 1). This represents a retention rate of ~89% from baseline to FU1 (1018/1145). Overall, the participants were on average 38 years old and were approximately 11.5 years on from their primary injury/ deployment. The majority (~90.4%) were Caucasian and held a lower rank. A greater proportion of the injured compared to the uninjured had left the military. Smoking and family history of cardiovascular disease were similar among those with and without CRTI.

Visceral fat mass (VFM) was significantly greater and RMSSD (HRV), compliance with WHO recommendations for LT-MVPA (physical activity) and 6MWD (physical function) were lower among the injured versus uninjured at baseline (Table 2). These differences were greatest among the injured amputees (Table 2). Hs-CRP was significantly greater among the injured amputees compared to the injured non-amputees and uninjured.

The mean SEVR at FU1 was lower in the injured (187.2, SD 39.7) compared to the uninjured (194.1, SD 31.5). We observed the SEVR to be 181.9 (SD 30.0) in the injured amputee sub-group.

Correlations between the mediators varied between 0.09 and 0.36 showing weak correlation (S1 Table).

### Visceral Fat Mass (VFM)

The association between injury and SEVR was mediated by VFM (Table 3). We observed a 5.8-unit total decrease in the SEVR for the injured compared to the uninjured (total effect). Of these, around 1.8 units are possibly due to VFM and around 4 units due to other mechanisms. The proportion mediated was 31%. The CDE, setting the mediator to the grand mean of 460 grams, was −4.32 (−8.61, −0.04); therefore, if all participants had a visceral fat measurement of 460 grams the injured would have a lower SEVR of 4.3 units compared to the uninjured. The proportion eliminated is 25.9%.

In the subgroup analysis, we observed a decrease in SEVR of 11.2 units for those with amputations compared to the uninjured I (Table 4). Of these, 4.4 may be explained by the increase in VFM (proportion mediated 37%). The CDE showed that in participants with amputations (setting the mediator to the grand mean) those who had a VFM of 460 grams on average had a lower SEVR of 7.29 (−13.2, −1.44) compared to the uninjured (Fig 2) with the proportion eliminated 34.7%. The results for the injured non amputees showed no direct, total or indirect effect.

**Table 1.  Demographics of all participants stratified by injury and amputation status at the first (three-year) follow up visit (n = 1018).**

| | Uninjured (n = 504) | Injured (n = 514) | Injured No Amputation (n = 374) | Injured Amputation (n = 140) |
|---|---|---|---|---|
| Age at Follow up 1 assessment, years | 38.3 (5.4) | 38.0 (5.3) | 38.3 (5.5) | 37.2 (5.4) |
| Time from deployment/injury to baseline assessment (years), median (IQR) | 7.7 (2.5) | 8.3 (2.9) | 8.6 (3.1) | 7.4 (2.2) |
| Time from deployment/injury to follow up assessment (years), median (IQR) | 11.2 (2.5) | 11.6 (2.8) | 11.8 (3.0) | 11.0 (2.1) |
| Left military service | 159 (31.7) | 407 (79.5) | 270 (72.6) | 137 (97.9) |
| Rank/NSEC (at sampling) | | | | |
| Junior | 298 (59.1) | 362 (70.4) | 251 (67.1) | 111 (79.3) |
| Senior | 136 (27.0) | 97 (18.9) | 80 (21.4) | 17 (12.1) |
| Officer | 70 (13.9) | 55 (10.7) | 43 (11.5) | 12 (8.6) |
| New Injury Severity Score, median (IQR) | – | 12 (5, 22) | 9 (4, 17) | 22 (14, 29) |
| Ethnicity | | | | |
| Caucasian | 456 (90.5) | 464 (90.3) | 337 (90.1) | 127 (90.7) |
| Other (including mixed race) | 48 (9.5) | 50 (9.7) | 37 (9.9) | 13 (9.3) |
| Family history of CVD§ | 117 (20.7) | 104 (18.0) | 70 (16.8) | 34 (21.1) |
| Smoking history at FU1 | | | | |
| Current smoker | 97 (19.3) | 70 (13.6) | 49 (13.1) | 21 (15.0) |
| Ex smoker | 176 (34.9) | 191 (37.2) | 137 (36.6) | 54 (38.6) |
| Never Smoked | 231 (45.8) | 253 (49.2) | 188 (50.3) | 65 (46.4) |

Data are presented as mean (SD) or n (%) unless otherwise stated.

§ First degree relative; IQR, inter-quartile range; NSEC, National Socioeconomic Classification of social status; CVD, Cardiovascular disease; IQR, inter-quartile range; FU1, follow up 1

**Table 2.  Differences in cardiovascular risk measures at baseline and the subendocardial viability ratio (SEVR) at the first follow up on average three years later.**

| | Uninjured (n = 504) | Injured (n = 514) | P value | Injured no amputation (n = 374) | Amputation (n = 140) | P value† |
|---|---|---|---|---|---|---|
| **Baseline** | | | | | | |
| Hs-CRP, mg/l, median (IQR) | 0.81 (1.11) | 1.00 (1.40) | 0.10 | 0.86 (1.27) | 1.20 (1.90) | <0.001 |
| RMSSD, ms, median (IQR) | 46 (38) | 40 (32.0) | <0.001 | 40 (31.0) | 38 (36.0) | 0.002 |
| Visceral fat mass, g | 440.5 (169.6) | 486.0 (200.8) | <0.001 | 474.9 (195.4) | 516.1 (212.5) | <0.001 |
| Six Minute Walk distance, m | 630.3 (95.5) | 571.8 (124.3) | <0.001 | 598.0 (113.9) | 494.3 (121.8) | <0.001 |
| Compliance with WHO recommended levels of LT-MVPA, n (%) | | | | | | |
| No | 175 (36.5) | 230 (46.9) | 0.001 | 162 (44.6) | 68 (53.5) | 0.001 |
| Yes | 305 (63.5) | 260 (53.1) | | 201 (55.4) | 59 (46.5) | |
| **Follow-up 1** | | | | | | |
| Subendocardial viability ratio | 194.1 (39.7) | 187.2 (31.5) | 0.002 | 189.2 (31.9) | 181.9 (30.0) | <0.001 |

Data are presented as mean (SD) unless otherwise stated.

Hs-CRP, high-sensitivity C reactive protein; RMSSD root mean square of successive differences; WHO, World Health Organization; LT-MVPA, leisure-time moderate or vigorous physical activity

†Overall p value for uninjured, injured no amputation and amputation.

**Table 3. Total, Direct and Indirect Effects for the association between CRTI and SEVR and mediators.**

| Mediator | N (%) | Total Effect coef (95% CI) | Natural Direct Effect coef(95% CI) | Natural Indirect Effect coef(95% CI) |
|---|---|---|---|---|
| Visceral Fat Mass | 1009 | −5.80 (−10.22, −1.38) | −4.00 (−8.41, 0.40) | −1.80 (−2.90, −0.67) |
| RMSSD | 765 | −4.74 (−9.60, 0.11) | −2.93 (−7.71, 1.86) | −1.82 (−3.45, −0.19) |
| Hs-CRP | 963 | −6.13 (−10.58, −1.68) | −5.84 (−10.24, −1.43) | −0.29 (−0.87, 0.28) |
| Six Minute Walk distance | 985 | −5.24 (−9.46, −1.01) | −3.45 (−7.95, 1.05) | −1.79 (−3.16, −0.41) |
| Non-Compliance with WHO recommended weekly LT-MVPA | 970 | −4.95 (−9.40, −0.49) | −4.54 (−8.99, −0.08) | −0.41 (−1.00, 0.18) |

CRTI, combat-related traumatic injury; SEVR, subendocardial viability ratio; Hs-CRP, high-sensitivity C reactive protein; RMSSD root mean square of successive differences; WHO, World Health Organization; LT-MVPA, leisure-time moderate or vigorous physical activity

**Table 4. Total, Direct and Indirect Effects for the association between amputation status and SEVR and mediators.**

| Mediator | Total Effect coef (95% CI) | Natural Direct Effect coef(95% CI) | Natural Indirect coef(95% CI) |
|---|---|---|---|
| Visceral Fat Mass | | | |
| Non amp vs Uninjured | −3.96 (−8.79, 0.88) | −2.96 (−7.76, 1.84) | −1.00 (−2.00, 0.01) |
| Amp vs Uninjured | −11.16 (−17.12, −5.21) | −6.73 (−12.80, −0.67) | −4.43 (−7.31, −1.56) |
| RMSSD | | | |
| Non amp vs Uninjured | −1.55 (−6.88, 3.76) | −0.14 (−5.31, 5.04) | −1.42 (−2.96, 0.14) |
| Amp vs Uninjured | −12.06 (−18.71, 5.41) | −7.70 (−14.03, −1.37) | −4.36 (−7.41, −1.30) |
| Hs-CRP | | | |
| Non amp vs Uninjured | −4.4 (−9.30, 0.45) | −4.45 (−9.28, 0.38) | 0.02 (−0.67, 0.71) |
| Amp vs Uninjured | −10.66 (−16.46, −4.86) | −10.27 (−16.11, −4.42) | −0.40 (−1.90, 1.11) |
| Six Minute Walk Test | | | |
| Non amp vs Uninjured | −3.52 (−8.23, 1.18) | −2.63 (−7.41, 2.15) | −0.89 (−1.93, 0.14) |
| Amp vs Uninjured | −10.41 (−16.67, −4.15) | −7.68 (−15.97, 0.62) | −2.73 (−7.87, 2.41) |
| Non-Compliance with WHO recommended weekly MVPA | | | |
| Non amp vs Uninjured | −3.29 (−8.05, 1.47) | −3.26 (−8.03, 1.50) | −0.02 (−0.50, 0.45) |
| Amp vs Uninjured | −9.77 (−15.97, −3.57) | −7.46 (−13.69, −1.23) | −2.31 (−4.44, −0.18) |

SEVR, subendocardial viability ratio; Hs-CRP, high-sensitivity C reactive protein; RMSSD root mean square of successive differences; WHO, World Health Organization; LT-MVPA, leisure-time moderate or vigorous physical activity

### Heart rate variability (RMSSD)

We also observed a mediating effect on the association by RMSSD (Table 3). Of the total effect of a 4.7-unit decrease, around 1.8 units are potentially due to RMSSD (proportion mediated 38%).

In the subgroup analysis, we observed a decrease in SEVR of 12 units for those with amputations compared to the uninjured (Table 4). Of these, over 4 are likely due to the decrease in RMSSD. The results for the injured non amputees showed no direct, total effect or indirect effect.

### Physical function and activity

We found evidence that the 6MWD, but not the LT-MVPA, mediated the association between CRTI and SEVR (Table 3). However, in the subgroup analysis, LT-MVPA mediated the association between those with amputations (vs uninjured) and SEVR (NIE −2.31 [−4.44, −0.18]). The total effect was −9.77 (−15.97, −3.57) (Table 4). The CDE, if the mediator was set so that every participant complied with MVPA, was −4.50 (−12.90, 3.91) and the proportion eliminated is 54%.

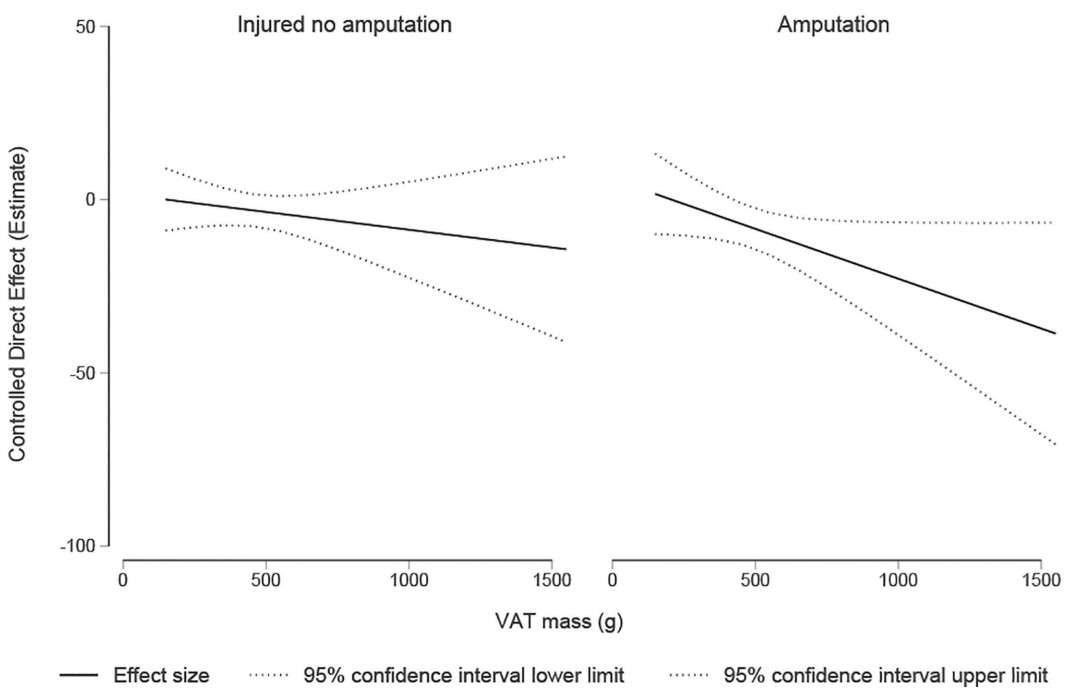

**Fig 2. Controlled direct effect of Injury (Uninjured(ref), injured non amputee and amputee) on SEVR (outcome) when visceral fat mass (g) (mediator) is fixed at values that range from the minimum to the maximum observed values.** Includes an exposure-mediator interaction.

### Inflammation (Hs-CRP)

We found no evidence that inflammation (Hs-CRP) mediated the association between injury and SEVR (Table 3).

### Discussion

This is the first observational cohort study to examine the potential mediating effects of cardiovascular risk factors on the relationship between CRTI and MBFR, measured using the SEVR. We observed that VFM, 6MWD and RMSSD (HRV) mediated the association between CRTI and SEVR. In the subgroup analysis, VFM, RMSSD and non-compliance with recommended LT-MPVA mediated the relationship between the injured amputation group (vs uninjured) and SEVR. Hs-CRP did not play a significant mediating role in the relationship between CRTI and SEVR.

In the baseline analysis of ADVANCE, we found that SEVR was significantly lower among the participants with CRTI compared with uninjured participants of similar age, sex, role and time of deployed [7]. It was noted that these differences were greater and the SEVR lower in participants with traumatic amputation and independent of age, rank, ethnicity and time from injury/deployment. Moreover, the lowest SEVR values were found in the participants with the most severe amputation injuries (multiple and above knee amputations). These findings raised important questions as to the potential causal relationship between CRTI and SEVR and the possible mediating factors in this relationship. A low SEVR is suggestive of an imbalance in myocardial oxygen supply-demand and is an increasingly validated cardiovascular risk measure [11,27]. The SEVR has been shown to be influenced by several cardiovascular risk factors including obesity, vascular inflammation, autonomic balance, exercise and fitness [11,27]. Reduced SEVR has been independently associated with adverse cardiovascular events and is strongly correlated with increased Hs-CRP and visceral fat and lower HRV and exercise burden [11,28]. Whist the SEVR readings in our study were generally in the normal range, they were 13 units lower in the injured amputees which is an effect size not dissimilar to that observed in a recent study of metabolic

syndrome among healthy adults and previously among patients with hypertension, coronary artery disease (and its severity) and diabetes mellitus compared with controls [11,29,30].

The selection of mediators that were examined in this study was based on strong scientific premises and further informed by the baseline analysis of the ADVANCE study. At baseline, we found that vascular inflammation (e.g., Hs-CRP) and abdominal obesity (e.g., abdominal waist circumference and visceral fat) were significantly greater and 6MWD, LT-MVPA and HRV were lower in participants with CRTI compared to the uninjured comparison group; these differences were even greater with worsening injury severity and among those with amputation [4,7]. Hs-CRP and visceral fat and lower 6MWD, LT-MVPA and HRV have been shown to be strongly related to elevated cardiovascular risk and adverse cardiovascular outcomes [31,32]. Current evidence has supported their causal relationship to atherosclerotic cardiovascular disease development [31]. This causal inference has been strengthened by data from randomised studies in which interventions such as exercise and diet/medication (e.g., statins and GLP-1 agonists) have led to reductions in Hs-CRP and visceral fat and increased autonomic function (HRV) and improvement in cardiovascular outcomes [33–35]. Hence, Hs-CRP, VFM, physical activity and HRV are modifiable cardiovascular risk markers.

There is a growing body of evidence to suggest that traumatic injury is associated with increased cardiovascular risk. [1,4,5]. Hallmarks of this increased risk profile include reduced physical function, recreational exercise, chronic low-grade vascular inflammation, relative obesity, autonomic imbalance and potentially dyslipidaemia. The mechanisms to explain these links are highly complex, poorly understood and influenced by the mechanism of injury (e.g., blast) and injury type (e.g., traumatic brain injury versus single lower limb amputation), pain, mental health and post traumatic growth [1,4,5].

Our finding of a mediating effect of visceral fat and autonomic imbalance on the relationship between CRTI and SEVR makes biological sense. Traumatic injury affects the hypothalamic-pituitary-adrenal axis to increase cortisol secretion, which is a well-recognised stimulus for visceral fat deposition and abdominal obesity. This may be accentuated in the presence of traumatic brain injury (TBI) which has been shown to affect 16.9% of the injured cohort in ADVANCE [36]. Reduced physical function and exercise are well-recognised sequelae of physical injury leading to sarcopenia, autonomic imbalance and fat gain. Its impact is likely to be even greater with traumatic amputation which is borne out by our current study. Traumatic injury can also lead to significant psychological (e.g., PTSD, anxiety, depression and insomnia), and behavioural changes (high calorific diets) that promote relative obesity and autonomic imbalance [37]. Visceral fat is associated with an increased risk of atherosclerosis compared to subcutaneous fat and is important source of proinflammatory adipokines. These circulating adipokines can target distant organs to effect body weight, energy storage, insulin sensitivity, glucose regulation which are an important determinant of coronary blood flow and myocardial perfusion [38].

HRV and visceral fat are important determinants of resilience and long-term health of military servicemen. Hence, interventional strategies to increase HRV and reduce visceral fat, could have important translational advantages in the recovery trajectory of combat injured servicemen. There are several plausible strategies that could be introduced to enhance post-injury recovery/rehabilitation pathways including a stronger emphasis on a healthier lifestyle (improved diet, sleep discipline and exercise) and autonomic training [39]. ADVANCE is currently designing a feasibility study examining the use of HRV biofeedback training as an intervention to improve mental and physical health in injured British servicemen.

Whilst compliance with WHO recommended levels of LT-MVPA and 6MWD were significantly lower with the injured versus uninjured, LT-MVPA was not a significant mediator to SEVR, except among the amputees versus the uninjured. There are several factors that might explain this. We used a binary (yes or no), but well-established WHO recommended cut-off for LT-MVPA, which represents the minimum burden of weekly exercise. Additionally, the CDE suggested, cautiously, that interventions on participants' physical activity – in which all participants were to comply with the WHO recommended level of exercise – could eliminate about half of the effect of amputation injury on SEVR. It is well-known that questionnaire data (from IPAQ and other exercise questionnaires) is subjective and prone to recall bias. Whilst we focussed on LT-MVPA to help reduce the potential recall bias, work-related or incidental activity can be substantial in some populations. MVPA does not account for individual responses in which different participants could achieve the minimum standard yet the total

duration of LT-MVPA, intensity and physiological challenges could be different. Finally, compliance with a physical activity recommendation is not a direct or robust measure of cardiorespiratory fitness which is known to influence the SEVR in healthy adults [40]. Nevertheless, in a recent meta-analysis of 661,137 Western adults, it was shown that compared to those reporting no LT-MVPA, those achieving the minimum WHO recommended standard for LT-MVPA was associated with a 31% lower all-cause mortality (hazard ratio 0.69, 0.67–0.70) [19].

We found that Hs-CRP was not a significant mediator in the relationship between CRTI and SEVR. Obesity, visceral fat, lower physical activity are all-well recognised drivers of chronic vascular inflammation and an increased Hs-CRP. Elevated Hs-CRP is associated with endothelial dysfunction, which impairs coronary microcirculation. Whilst there is evidence from both observational and interventional studies to support a significant inverse association between inflammation and myocardial blood flow, this work largely relates to systemic inflammatory conditions (e.g., rheumatoid arthritis) and other pro-inflammatory cytokines such as Galectin-3 [41]. Moreover, we cannot exclude the possibility that these temporal effects may take time to be fully appreciated and hence it may be too early to demonstrate this relationship, longer term follow up of the cohort will be required. Conversely, we cannot dispute a potentially mitigating and positive effect of education, psychological support, health promotion and rehabilitation related to involvement in the ADVANCE study itself.

This study has additional strengths and weaknesses/limitations that need highlighting. One of the main strengths is that we have examined a well-characterised cohort of injured and uninjured combat deployed military personnel who were matched on age, occupational role, deployment location and period at the point of selection. All study measures were obtained under strict research conditions. The ADVANCE study is the only ongoing prospective cohort study examining the effects of CRTI on physical and mental health outcomes. Our examination of established cardiovascular risk markers and the longitudinal design of our study with an average time of 11.5 years post injury/deployment, are further strengths. Finally, our sample size is relatively large given the unique nature of the study population and our retention rate of ~89% from baseline to the three year follow up is excellent. In terms of limitations, we cannot rule out any unmeasured confounding which may bias the estimates. There are potential confounders (e.g., smoking, heartrate, blood pressure) of the mediator-outcome relationship, [11]. however, as these variables were likely on the casual pathway from injury to outcome, they were not included in the analysis. Therefore, the strong assumptions of no unmeasured mediator-outcome confounders may not hold. Additionally, estimating mediators separately can be misleading and the proportion mediated should not be summed to get an overall proportion mediated due to possible correlation between mediators. Whilst there were some weak correlations between mediators, most showed no or very weak correlation. Data were missing on RMSSD for 25% of the participants. This was due to non- availability of the raw pulse-waveform data at baseline which is crucial for HRV analysis. It is noteworthy that HRV remains significantly lower among the injured versus uninjured using full cohort data at FU1 (*in press*). Finally, we used a non-invasive method of SEVR measurement. This method has the advantage of being more practical, cost-effective and repeatable than the traditional invasive method requiring invasive haemodynamic assessment.

In conclusion, we found that visceral fat, HRV and physical function, measured using 6MWD, mediated the relationship between CRTI and SEVR. LT-MVPA was only a significant mediator among those injured with amputations. In those with amputations, future interventions to increase fitness and reduce visceral fat could result in a reduced effect of injury on SEVR. Longer-term follow-up data from the ADVANCE Study is needed to robustly determine the temporal effects of this relationship and the prospects for an interventional study to mitigate these potential adverse cardiovascular effects.

## Supporting information

**S1 Table. Correlations (ρ) between mediators.**
(DOCX)

## Acknowledgments

We wish to thank all the research staff at Stanford Hall who helped with the ADVANCE study, including Emma Coady, Grace Blissitt, Melanie Chesnokov, Daniel Dyball, Sarah Evans, Guy Fraser, Nicola Goodman, Alison Hever, Meliha Kaya-Barge, Jocelyn Keshet-Price, Eleanor Miller, Steven Parkes, Bharti Patel, Samantha Paul, David Pernet, Vlad Pop, Helen Prentice, Ursula Pucilowska, Stefan Sprinckmoller, Jodie Stevenson, Lalji Varsani, Anna Verey, Molly Waldron, Owen Walker, Farheen Dairkee, Tasarla White, Seamus Wilson, and Severija Juškaitė.

## Author contributions

**Conceptualization:** Christopher J Boos, Susie Schofield, Rabeea Maqsood, Anthony MJ Bull, Nicola T Fear, Paul Cullinan, Alexander N Bennett.

**Data curation:** Christopher J Boos, Rabeea Maqsood, Alexander N Bennett.

**Formal analysis:** Susie Schofield.

**Funding acquisition:** Anthony MJ Bull, Nicola T Fear, Paul Cullinan, Alexander N Bennett.

**Investigation:** Christopher J Boos, Susie Schofield, Rabeea Maqsood, Anthony MJ Bull, Nicola T Fear, Paul Cullinan, Alexander N Bennett.

**Methodology:** Christopher J Boos, Susie Schofield, Rabeea Maqsood, Anthony MJ Bull, Nicola T Fear, Paul Cullinan, Harriet Kemp, Alexander N Bennett.

**Project administration:** Christopher J Boos, Anthony MJ Bull, Nicola T Fear, Paul Cullinan, Harriet Kemp, Alexander N Bennett.

**Resources:** Christopher J Boos, Anthony MJ Bull, Paul Cullinan, Alexander N Bennett.

**Software:** Susie Schofield, Rabeea Maqsood.

**Supervision:** Christopher J Boos, Anthony MJ Bull, Alexander N Bennett.

**Validation:** Christopher J Boos, Susie Schofield.

**Visualization:** Christopher J Boos.

**Writing – original draft:** Christopher J Boos, Susie Schofield.

**Writing – review & editing:** Christopher J Boos, Susie Schofield, Rabeea Maqsood, Anthony MJ Bull, Nicola T Fear, Paul Cullinan, Harriet Kemp, Alexander N Bennett.

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
