## [Decision Letter · Decision Letter 0]

20 Nov 2025

Dear Dr. Boos,

We look forward to receiving your revised manuscript.

Kind regards,

Niema M. Pahlevan, PhD

Academic Editor

PLOS ONE

Journal Requirements:

2. Thank you for providing the following Funding Statement:

“This study is a project by the ADVANCE study (grant number: ADV-ADMR-03), which is funded through the ADVANCE Charity. Key contributors to the charity are the Headley Court Charity (principal funder), HM Treasury (LIBOR Grant), Help for Heroes, Nuffield Trust for the Forces of the Crown, Forces in Mind Trust, National Lottery Community Fund, Blesma - The Limbless Veterans, the UK Ministry of Defence, and the Office for Veterans’ Affairs (OVA). The funders of the study had no role in study design, data collection, data analysis, data interpretation, or writing of the manuscript.”

We note that one or more of the authors is affiliated with the funding organization, indicating the funder may have had some role in the design, data collection, analysis or preparation of your manuscript for publication; in other words, the funder played an indirect role through the participation of the co-authors.

If the funding organization did not play a role in the study design, data collection and analysis, decision to publish, or preparation of the manuscript and only provided financial support in the form of authors' salaries and/or research materials, please review your statements relating to the author contributions, and ensure you have specifically and accurately indicated the role(s) that these authors had in your study in the Author Contributions section of the online submission form. Please make any necessary amendments directly within this section of the online submission form.  Please also update your Funding Statement to include the following statement: “The funder provided support in the form of salaries for authors [insert relevant initials], but did not have any additional role in the study design, data collection and analysis, decision to publish, or preparation of the manuscript. The specific roles of these authors are articulated in the ‘author contributions’ section.”

If the funding organization did have an additional role, please state and explain that role within your Funding Statement.

Please also provide an updated Competing Interests Statement declaring this commercial affiliation along with any other relevant declarations relating to employment, consultancy, patents, products in development, or marketed products, etc.

4. Please note that your Data Availability Statement is currently missing [the repository name and/or the DOI/accession number of each dataset OR a direct link to access each database]. If your manuscript is accepted for publication, you will be asked to provide these details on a very short timeline. We therefore suggest that you provide this information now, though we will not hold up the peer review process if you are unable.

5. Please be informed that funding information should not appear in the Acknowledgments section or other areas of your manuscript. We will only publish funding information present in the Funding Statement section of the online submission form. Please remove any funding-related text from the manuscript.

7. We note you have included a table to which you do not refer in the text of your manuscript. Please ensure that you refer to Table 4 in your text; if accepted, production will need this reference to link the reader to the Table.

Reviewers' comments:

Reviewer's Responses to Questions

**Comments to the Author**

1. Is the manuscript technically sound, and do the data support the conclusions?

Reviewer #1: Yes

Reviewer #2: Yes

Reviewer #3: Yes

2. Has the statistical analysis been performed appropriately and rigorously?

Reviewer #1: Yes

Reviewer #2: Yes

Reviewer #3: Yes

3. Have the authors made all data underlying the findings in their manuscript fully available?

Reviewer #1: Yes

Reviewer #2: No

Reviewer #3: Yes

4. Is the manuscript presented in an intelligible fashion and written in standard English?

Reviewer #1: Yes

Reviewer #2: Yes

Reviewer #3: Yes

Reviewer #1: This is an original paper that examines the relationship between certain cardiovascular risk factors including visceral fat, heart rate variability, physical functioning in patients with combat related traumatic injury or without injury or amputation and subendocardial viability ratio. They find reduced SEVR in injured amputees and other fascinating associations. Some of this seems like common sense. For example, an amputee is not going to be able to necessarily exercise in the same fashion as a non injured amputee and therefore coronary reserve is likely to be better in the non injured individual. My major criticism is that we are not provided information about other important cardiovascular risk factors here: Cholesterol, LDL, HDL, triglyceride values; information on diabetes, Hemoglobin A1C, hypertension, medicines. For the HRV measures how long was the pulse or ECG measured? How were arrhythmias handled? Data on depression/anxiety and emotional stress are not provided, but it is well known that psychological stress can contribute to cardiovascular events. What medicines were the participants on .... were any on vasodilators including things like calcium channel blockers, nitrates, PDE5 inhibitors... Viagra like drugs, since these can all cause vasodilation including of the coronary arteries. There should be some discussions related to these issues or data provided if available on medicines.

Reviewer #2: The authors present a valuable analysis using the ADVANCE cohort to investigate the mediating pathways linking combat-related traumatic injury (CRTI) to long-term reductions in myocardial blood flow reserve (MBFR), as measured by the Subendocardial Viability Ratio (SEVR). The study addresses a clinically important question regarding the long-term cardiovascular health of service personnel. The use of a robust, prospective cohort and the application of mediation analysis are strong methodological points. The manuscript is generally well-structured. However, several critical details concerning the mediation model's reporting, methodology, and interpretation need to be fully clarified and addressed before moving forward.

Major Comments:

1- The core methodology of this manuscript is mediation analysis. It is crucial to report the analysis with explicit detail. Given that the cohort was frequency-matched (by age, rank, and deployment), please detail how these matching factors were included in the mediation models (as covariates or via a more specialized approach).

2- The results must clearly state the full statistical picture of the mediation. What was the calculated Indirect Effect (IE) through the identified mediator(s)? Explicitly state whether the finding represents full mediation (CDE non-significant) or partial mediation (CDE significant). This is a required step for interpreting mediation models.

3- The legend for Figure 2 mentions an exposure-mediator interaction. If this interaction term was statistically significant, the standard interpretation of a simple indirect effect is no longer valid. The analysis must then focus on Conditional Indirect Effects (i.e., how the mediation effect changes at different levels of the moderator, which here appears to be the mediator). Please elaborate on this analysis.

4- Did the authors run separate single-mediator models for each factor, or did they perform a parallel or serial multiple mediation analysis? If separate models were run, please justify this choice and explain how the issue of Type I error (false positives) from running multiple tests was addressed.

5- Which version of SEVR was used? Please explain if LV pressure was not used in the limitations.

6- SEVR is a dynamic measure and highly sensitive, which can be influenced by the mediators themselves (e.g., fitness via 6MWD, autonomic control via RMMSD). Please confirm that the SEVR measurements were standardized (e.g., performed under the same rest/fasting conditions) and specify if blood pressure and heart rate at the time of SEVR measurement were included as time-varying covariates in the outcome model. If not, this should be discussed as a limitation.

7- The authors mentioned sex (male) in the methods once. Did the study only include male servicemen? Did you also include females? If yes, what was any sex-difference observed in mediator analysis?

Minor Comments:

1- Given that the mediators were assessed approximately eight years post-injury/deployment, and the outcome (SEVR) was measured later, there is a risk of reverse causality or confounding due to post-injury lifestyle changes. Please include a dedicated limitations section addressing the temporal relationship and the challenge of establishing true causality in this context.

2- Please confirm and state whether the analysis includes all 1018 servicemen mentioned, or if there was subsequent drop-out or missing data for the SEVR measurement. If the sample size is smaller for the final analysis, please state the N used in the Methods/Results.

Reviewer #3: Please see the full review in the attached PDF.

Summary:

This manuscript entitled “Mediating Factors in the Relationship between Combat-related Traumatic Injury and Myocardial Blood Flow Reserve: the ADVANCE Cohort Study” investigates whether specific cardiovascular risk factors mediate the relationship between combat-related traumatic injury (CRTI) and myocardial blood flow reserve, represented by the subendocardial viability ratio (SEVR), using longitudinal data from the ADVANCE cohort. The study leverages a valuable dataset of UK servicemen (n=1,018, 504 with CRTI, of whom 140 were amputees, and 514 uninjured men, frequency-matched, by age, rank and deployment) with a follow-up duration for more than a decade after deployment. From the standpoint of public health significance, innovation, and scientific contribution, this paper represents a substantive and meaningful addition to the emerging literature on the long-term cardiovascular consequences of military trauma. The manuscript is suitable for publication pending modifications and clarifications described below:

Significance and Impact

The primary strength of this work lies in addressing an understudied question with substantial clinical implications: why combat-injured veterans exhibit elevated cardiovascular risk even years after injury. While prior papers from the same ADVANCE cohort established associations between CRTI, adverse cardiometabolic profiles, and reduced SEVR, this study advances the field by investigating the underlying mechanisms. Understanding modifiable mediators such as visceral fat mass, heart-rate variability, and physical function is crucial for designing targeted interventions and informing long-term rehabilitation strategies for injured service members. Given the large population of veterans with blast injuries, limb loss, and related trauma, these insights could have high translational relevance for both defense health systems and civilian trauma rehabilitation.

In terms of the approach, this study’s main novelty lies in its rigorous application of counterfactual mediation analysis. Prior literature linking traumatic injury to cardiovascular risk has largely been retrospective, cross-sectional, or limited to mortality outcomes. By contrast, to strengthen causal inference, this manuscript uses longitudinal measurements: baseline (~8 years post-injury) for mediators and follow-up (~3 years later) for SEVR. The approach allows the authors to quantify the extent to which specific physiological domains (autonomic balance, adiposity, inflammation, physical activity, and physical function) explain the CRTI–SEVR association. The subgroup analysis focusing on traumatic amputation is particularly innovative and important, offering mechanistic hypotheses tailored to a clinical subgroup known to be at heightened cardiovascular risk.

**Do you want your identity to be public for this peer review?** For information about this choice, including consent withdrawal, please see our Privacy Policy

Reviewer #1: No

Reviewer #2: **Yes:** Rashid Alavi

Reviewer #3: No

---

## [Author Response · Author response to Decision Letter 1]

28 Jan 2026

We would like to sincerely thank the three reviewers for taking the time to examine our manuscript and to provide such comprehensive reviews. We have responded (R) to each of the comments raised. We feel that we have been able to address all of the comments raised. We have, as requested also included both a tracked changes file and one without showing the tracked changes. We hope that our manuscript might now be considered suitable for publication in PLOS One.

Reviewer #1: This is an original paper that examines the relationship between certain cardiovascular risk factors including visceral fat, heart rate variability, physical functioning in patients with combat related traumatic injury or without injury or amputation and subendocardial viability ratio. They find reduced SEVR in injured amputees and other fascinating associations. Some of this seems like common sense. For example, an amputee is not going to be able to necessarily exercise in the same fashion as a non injured amputee and therefore coronary reserve is likely to be better in the non injured individual. My major criticism is that we are not provided information about other important cardiovascular risk factors here: Cholesterol, LDL, HDL, triglyceride values; information on diabetes, Hemoglobin A1C, hypertension, medicines.

R. Thank you for raising this very important point. In our study we examined the examined the mediating role of cardiovascular risk factors, shown to significantly greater with CRTI at study baseline (~8 years post-injury/deployment) on SEVR, measured three-years later (FU1).

There have been several Cardiovascular manuscripts that have been published in relation to the baseline data collection from the full ADVANCE (n=1144) Cohort. And design and protocol of the ADVANCE study have been published and cited in our current manuscript.

These Cardiovascular manuscripts have examined an extremely wide range of cardiovascular risk factors among the full cohort of 579 male adult UK combat veterans (UK-Afghanistan War 2003-2014) with combat injury CRTI who were frequency-matched to 565 uninjured men by age, service, rank, regiment, deployment period and role-in-theatre. Our baseline publications have also examined the Influence of injury type (amputation), severity (Injury severity scores) and mechanism (blast etc). The comparative risk factors examined at baseline (between the injured and uninjured) have included measures of inflammation (Hs-CRP, neutrophil-lymphocyte ratio), Arterial stiffness (arterial augmentation index and pulse wave velocity), Cardiometabolic health (metabolic syndrome), glycaemic status (fasting glucose and HbA1C), obesity (BMI, abdominal waist circumference), physical activity (using the IPAQ questionnaire) and physical function (6MWD) measures of visceral fat (using DEXA), fasting blood lipid profiles, cardiovascular proteomics (in submission), heart rate variability (HRV) and non-invasive measurement of myocardial blood flow (SEVR using the Vicorder device) as well as the comparative global cardiovascular risk prediction scores using the 21-item QRISK III Scoring tool. We have also examined the influence of a variety of metal health assessments including PTSD, anxiety, depression and sleep on various measures of cardiovascular risk including HRV (measured at baseline). Of note the SEVR was significantly lower in the injured vs uninjured group at baseline and even lower in those with more severe injuries.

Several examples of these publications in relations to the baseline data collection for ADVANCE are displayed below.

1. Boos CJ, Schofield S, Cullinan P, Dyball D, Fear NT, Bull AMJ, Pernet D, Bennett AN; ADVANCE study. Association between combat-related traumatic injury and cardiovascular risk. Heart. 2022 Mar;108(5):367-374. doi: 10.1136/heartjnl-2021-320296. Epub 2021 Nov 25. PMID: 34824088; PMCID: PMC8862100.

2. Boos CJ, Schofield S, Bull AMJ, Fear NT, Cullinan P, Bennett AN; ADVANCE Study. The relationship between combat-related traumatic amputation and subclinical cardiovascular risk. Int J Cardiol. 2023 Nov 1;390:131227. doi: 10.1016/j.ijcard.2023.131227. Epub 2023 Jul 30. PMID: 37527753.

3. Boos CJ, Haling U, Schofield S, Cullinan P, Bull AMJ, Fear NT, Bennett AN; ADVANCE Study. Relationship between combat-related traumatic injury and its severity to predicted cardiovascular disease risk: ADVANCE cohort study. BMC Cardiovasc Disord. 2023 Nov 27;23(1):581. doi: 10.1186/s12872-023-03605-0. PMID: 38012542; PMCID: PMC10680223.

4. Maqsood R, Schofield S, Bennett AN, Bull AM, Fear NT, Cullinan P, Khattab A, Boos CJ; ADVANCE study. Relationship between combat-related traumatic injury and ultrashort term heart rate variability in a UK military cohort: findings from the ADVANCE study. BMJ Mil Health. 2024 Dec 11;170(e2):e122-e127. doi: 10.1136/military-2022-002316. PMID: 36990509; PMCID: PMC11672064.

5. Dyball D, Bennett AN, Schofield S, Cullinan P, Boos CJ, Bull AMJ, Stevelink SA, Fear NT;

ADVANCE Study. The underlying mechanisms by which PTSD symptoms are associated with cardiovascular health in male UK military personnel: The ADVANCE cohort study. J Psychiatr Res. 2023 Mar;159:87-96. doi: 10.1016/j.jpsychires.2023.01.010. Epub 2023 Jan 17. PMID: 36696788.

6. Maqsood R, Schofield S, Bennett AN, Khattab A, Clark C, Bull AMJ, Fear NT, Boos CJ. The Influence of Physical and Mental Health Mediators on the Relationship Between Combat-Related Traumatic Injury and Ultra-Short-Term Heart Rate Variability in a U.K. Military Cohort: A Structural Equation Modeling Approach. Mil Med. 2024 Feb 27;189(3-4):e758-e765. doi: 10.1093/milmed/usad341. PMID: 37656495; PMCID: PMC10898941.

7. Maqsood R, Schofield S, Bennett AN, Khattab A, Bull AMJ, Fear NT, Cullinan P, Boos CJ. The inverse and non-linear association between central augmentation index and heart rate variability in a cohort of male British combat personnel- findings from the ADVANCE study. Blood Press. 2025 Dec;34(1):2524409. doi: 10.1080/08037051.2025.2524409. Epub 2025 Jul 10. PMID: 40549417.

Hence, it would have been inappropriate and repetitious to examine many of these cardiovascular risk markers again among the injured and uninjured at baseline. Moreover, for this current study we examined only the participants who went on to have SEVR measured again at follow up 1 three years later (approximately 11 years post injury/deployment) so that we could undertake a robust mediation model to investigate several of the potential factors that might explain the lower SEVR among the injured. This would allow us to potentially infer causation rather than simple association.

For the HRV measures how long was the pulse or ECG measured?

R. We had referenced the full details and methodology of the baseline HRV measurement in the manuscript. Essentially, this consisted of an ultrashort recording period of approximately 14 seconds. The calculation of the inter-beat interval to quantify HRV was obtained from arterial waveform data captured and stored during the measurement of pulse waveform velocity using the Vicorder device. We had full disclosure of the waveforms and any erroneous waveforms within the 14 second sequence were manually excluded during HRV signal processing in Kubios. This waveform HRV measurement method has been previously validated and published for our cohort. At the first follow up (follow up 1) for ADVANCE we introduced single lead ECG-recording in addition to the ongoing use of the Vicorder device to obtain pulse waveform analyses, arterial stiffness and central blood pressure measurements. Hence from follow up onwards (not baseline) we have the ability to measure HRV to gold standard using a continuous ECG method over 5-minute epochs. Nevertheless, we have found a strong agreement between the two methods of measuring HRV and this work has been published:

Maqsood R, Schofield S, Bennett AN, Khattab A, Bull AMJ, Fear NT, Boos CJ. Validity of Ultra-Short-Term Heart Rate Variability Derived from Femoral Arterial Pulse Waveform in a British Military Cohort. Appl Psychophysiol Biofeedback. 2024 Dec;49(4):619-627. doi: 10.1007/s10484-024-09652-3. Epub 2024 Jul 11. PMID: 38990252; PMCID: PMC11588943.

In the light of your comments we have updated the methods in our paper to make it clearer as to how HRV was measured and processed.

How were arrhythmias handled?

Thank you for raising this important issue. As HRV was measured at baseline using pulse waveform analysis and inter-beat interval (not RR interval-ECG) data we could not directly identify arrhythmias. The capacity to measure HRV using ECG data only became available at Follow up 1 onwards. However, we have full disclosure of the arterial waveform data and where there was clear evidence of extreme shortening of the consecutive arterial waveforms to suggest an ectopic with differing amplitudes (which were extremely rare) suggestive of potential ectopy this segment was voided from the HRV analysis as they were flagged in the Kubios HRB analysis software. Examination of the complete ECG data from FU1 have shown that the data is of high quality with no sustained arrhythmias seen.

Data on depression/anxiety and emotional stress are not provided, but it is well known that psychological stress can contribute to cardiovascular events. What medicines were the participants on .... were any on vasodilators including things like calcium channel blockers, nitrates, PDE5 inhibitors... Viagra like drugs, since these can all cause vasodilation including of the coronary arteries. There should be some discussions related to these issues or data provided if available on medicines.

R. Thank you for raising this very important point. The mental health outcomes for the full baseline ADVANCE Cohort have been published.

Dyball D, Bennett AN, Schofield S, Cullinan P, Boos CJ, Bull AMJ, Wessely S, Stevelink SAM, Fear NT; ADVANCE study. Mental health outcomes of male UK military personnel deployed to Afghanistan and the role of combat injury: analysis of baseline data from the ADVANCE cohort study. Lancet Psychiatry. 2022 Jul;9(7):547-554. doi: 10.1016/S2215-0366(22)00112-2. PMID: 35717965.

It was shown that whilst the injured had significantly worse mental health outcomes than the uninjured this was mainly driven by the injured non-amputees. The amputees actually had significantly lower anxiety, depression and PTSDs scores than the injured non amputees. Overall, sustaining a combat injury was associated with a 46–67% increase in odds of reporting PTSD, depression, and anxiety symptoms compared with uninjured personnel. Planned subgroup analysis results had shown these differences were driven mostly by those with non-amputation injuries, and that individuals who had amputation injuries had minimal differences in the odds of reporting probable PTSD, anxiety, or depression compared with uninjured personnel and notably lower odds of reporting poor mental health outcomes compared with individuals who had non-amputation-related injuries.

The primary outcome of interest in this current study is the subendocardial viability ratio (SEVR). An association between PTSD and mental health outcomes and SEVR has not been previously demonstrated. Hence, we aimed to investigate mediators that have been shown previously and form our baseline ADVANCE papers to be associated with the SEVR and potentially causally related.

Our group have also published a follow up study from the baseline ADVANCE cohort in which the relationship between PTSD clusters and cardiovascular risk were examined. In this second manuscript (see below) no associations were observed between PTSD symptom clusters and high sensitivity c-reactive protein, diastolic BP, total cholesterol or fasting glucose. PTSD were associated with increased cardiovascular risk via cardiometabolic (visceral fat and triglycerides) and haemodynamic functioning (heart rate) mechanisms, but not inflammation. Hence it is not unreasonable to hypothesise that PTSD could influence SEVR. However, this well beyond the scope of the current study and as previously mentioned PTSD was not worsened by amputee status.

Dyball D, Bennett AN, Schofield S, Cullinan P, Boos CJ, Bull AMJ, Stevelink SA, Fear NT; ADVANCE Study. The underlying mechanisms by which PTSD symptoms are associated with cardiovascular health in male UK military personnel: The ADVANCE cohort study. J Psychiatr Res. 2023 Mar;159:87-96. doi: 10.1016/j.jpsychires.2023.01.010. Epub 2023 Jan 17. PMID: 36696788.

Reviewer #2: The authors present a valuable analysis using the ADVANCE cohort to investigate the mediating pathways linking combat-related traumatic injury (CRTI) to long-term reductions in myocardial blood flow reserve (MBFR), as measured by the Subendocardial Viability Ratio (SEVR). The study addresses a clinically important question regarding the long-term cardiovascular health of service personnel. The use of a robust, prospective cohort and the application of mediation analysis are strong methodological points. The manuscript is generally well-structured. However, several critical details concerning the mediation model's reporting, methodology, and interpretation need to be fully clarified and addressed before moving forward.

Major Comments:

1- The core methodology of this manuscript is mediation analysis. It is crucial to report the analysis with explicit detail. Given that the cohort was frequency-matched (by age, rank, and deployment), please detail how these matching factors were included in the mediation models (as covariates or via a more specialized approach).

R. Apologies if this was not clear. We have now amended to text to show that age and rank were included as confounders in the mediation model. As the cohort was frequency matched on the exposure, no specialized methods were required.

“Confounders (age at injury/deployment and rank) of the exposure-mediator and mediator-outcome relationships were included in the models.”

2- The results must clearly state the full statistical picture of the mediation. What was the calculated Indirect Effect (IE) through the identified mediator(s)? Explicitly state whether the finding represents full mediation (CDE non-significant) or partial mediation (CDE significant). This is a required step for interpreting mediation models.

R. Thank you for your comment. We state the effect size and CI of the Natural Indirect Effect (NIE) for each mediator in the table. In the text we refer to the NIE, however we do focus more on the CDE and proportion eliminated due to the more relaxed assumptions of these and we do discuss the possible bias of the NIE due to possible exposure-induced confounders.

Discussing mediation in terms of full and partial was discussed, however more recently many authors have advised against this terminology.

Memon, Mumtaz & Cheah, Jun-Hwa & Ramayah, T. & Ting, Hiram & Chuah, Francis. (2018). Mediation Analysis: Issues and Recommendations. Journal of Applied Structural Equation Modeling. 2. i-ix. 10.47263/JASEM.2(1)01.

If we state that we have full mediation, it may also imply that we have explained the association between exposure and outcome. This of course would be misleading, especially as we have addressed our limitations of exposure-induced confounders. Therefore, rather than using the terms full and partial mediation, we provide the effects sizes and confidence intervals of the NIE, NDE and TE for the reader to draw conclusions.

3- The legend for Figure 2 mentions an exposure-mediator interaction. If this interaction term was statistically significant, the standard interpretation of a simple indirect effect is no longer valid. The analysis must then focus on Conditional Indirect Effects (i.e., how the mediation effect changes at different levels of the moderator, which here appears to be the mediator). Please elaborate on this analysis.

R. Thank you for your comment. If we were using the traditional methods of mediation (i.e Baron and Kenny) then, yes, our indirect estimates would be biased, as traditional methods are unable to handle interactions. However, utilising the counterfactual framework allows for the decomposition and identification of Direct and Indirect Effects in the presence of a mediator-outcome interaction (ref).

4- Did the authors run separate single

---

## [Decision Letter · Decision Letter 1]

2 Mar 2026

Mediating Factors in the Relationship between Combat-related Traumatic Injury and Myocardial Blood Flow Reserve: the ADVANCE Cohort Study

PONE-D-25-50727R1

Dear Dr. Boos,

We’re pleased to inform you that your manuscript has been judged scientifically suitable for publication and will be formally accepted for publication once it meets all outstanding technical requirements.

Kind regards,

Niema M. Pahlevan, PhD, FAHA

Academic Editor

PLOS One

Additional Editor Comments (optional):

Reviewers' comments:

Reviewer's Responses to Questions

**Comments to the Author**

Reviewer #2: All comments have been addressed

Reviewer #3: All comments have been addressed

2. Is the manuscript technically sound, and do the data support the conclusions?

Reviewer #2: Yes

Reviewer #3: Yes

3. Has the statistical analysis been performed appropriately and rigorously?

Reviewer #2: Yes

Reviewer #3: Yes

4. Have the authors made all data underlying the findings in their manuscript fully available?

Reviewer #2: Yes

Reviewer #3: (No Response)

5. Is the manuscript presented in an intelligible fashion and written in standard English?

Reviewer #2: Yes

Reviewer #3: Yes

Reviewer #2: All my comments have been addressed or discussed as limitations. No further comments. The manuscript can be published now.

Reviewer #3: (No Response)

**Do you want your identity to be public for this peer review?** For information about this choice, including consent withdrawal, please see our Privacy Policy

Reviewer #2: No

Reviewer #3: No

---

## [Editor Report · Acceptance letter]

PONE-D-25-50727R1

PLOS One

Dear Dr. Boos,

I'm pleased to inform you that your manuscript has been deemed suitable for publication in PLOS One. Congratulations! Your manuscript is now being handed over to our production team.

Kind regards,

on behalf of

Dr. Niema M. Pahlevan

Academic Editor

PLOS One